# Preparation and Tribological Properties of Lanthanum Stearate Modified Lubricating Oil for Wire Rope in a Mine Hoist

**DOI:** 10.3390/ma14195821

**Published:** 2021-10-05

**Authors:** Yewei Zhang, Qing Zhang, Yuxing Peng, Chen Wang, Xiangdong Chang, Guoan Chen

**Affiliations:** 1School of Mechanical and Electrical Engineering, Jiangsu Key Laboratory of Mine Mechanical and Electrical Equipment, China University of Mining and Technology, Xuzhou 221116, China; zhangyewei2015@163.com (Y.Z.); pengyuxing@cumt.edu.cn (Y.P.); wangchen@cumt.edu.cn (C.W.); changxiangdong@cumt.edu.cn (X.C.); 2Jiangsu Collaborative Innovation Center of Intelligent Mining Equipment, Xuzhou 221008, China; 3College of Mechanical and Electrical Engineering, Zaozhuang University, Zaozhuang 277160, China; 4Command Academy of the Corps of Engineers, Xuzhou 221004, China; cga0608@163.com

**Keywords:** hoisting wire rope, lanthanum stearate, lubricant, dispersion stability, tribological performance

## Abstract

In view of the serious friction and wear on the surface of a hoisting wire rope caused by the failure of lubrication under severe hoisting conditions, a study on the tribological characteristics of lanthanum stearate modified lubricating oil (LSMLO) was carried out. First, lanthanum stearate was prepared by the saponification reaction, and its surface morphology, chemical structure, thermal stability, and dispersion stability in IRIS-550A lubricating oil (IRIS) for wire rope were analyzed. Then, the tribological properties of LSMLO were investigated through four-ball friction tests and sliding wear tests of wire ropes. The results show that stearic acid almost completely reacts to produce lanthanum stearate, which has good thermal stability and a disordered layered structure. With the help of oleic acid, the dispersion stability of lanthanum stearate in IRIS can be significantly improved. The four-ball friction tests show that the optimal addition amount of lanthanum stearate in IRIS is 0.2 wt.%, and the CoF and wear scar diameter are reduced by about 35% and 25% respectively when lubricated with LSMLO compared to that with IRIS. LSMLO can better reduce the wear of the wire rope under different sliding speeds and contact loads than IRIS, and it exhibits improved anti-friction and anti-wear properties under high speed and low load.

## 1. Introduction

A mine hoist is a pharynx equipment connecting the ground and underground in the process of coal mining, and the hoisting wire rope is the only traction part connecting the hoisting container, so its performance directly determines the hoisting safety. With the increasing demand for deep coal resource mining, mine hoists are gradually developing in the direction of ultra-deep and heavy load. The multi-layer winding hoist has become the first choice for ultra-deep mining because of its small drum size and excellent lifting capacity. However, in the process of multi-layer winding and hoisting, poor or failure of lubrication between wire ropes on the drum will inevitably lead to direct contact wear and reduce their mechanical properties. In addition, fluctuations in load, speed, and acceleration will accelerate the wear of the wire ropes, thus incurring real potential safety hazards. In the past, scholars have done a lot of work on the friction, wear, and mechanical properties of wire ropes and wires.

Fretting wear is inevitable due to the difference of tensile deformation between steel wires during the use of wire rope. In order to explore the fretting wear characteristics of steel wires under tension, torsion, and both, Wang et al. [1,2] compared and analyzed the wear states of steel wires under the three conditions with the help of a test bench and found that the wear mark size and wear rate are the largest under the combined condition of extension and torsion. Xu et al. [3] set up a steel wire fretting wear test rig considering the convex–concave structure of the spiral contact between the steel wires, and the fretting wear tests between the spiral contact steel wires with different sliding amplitudes while contact loads under the action of tension–torsion coupling force were carried out. They found that the wear degree of steel wire increases with increasing sliding amplitude and contact load. Subsequently, Xu et al. [4] analyzed the influence of diameter and cross angle on fretting wear behavior between steel wires under the action of tension–torsion coupling force and found that the larger cross angle makes the contact area of steel wires easier to enter the local slip state. In order to explore the specific parameters affecting the fretting wear between steel wires, Kumar et al. [5] systematically carried out a large number of analysis tests, and concluded that the material characteristics, structure, fatigue, contact. and lubrication all affect the fretting wear between steel wires. The mechanical properties of steel wire directly determine the bearing capacity of rope, so some scholars have conducted a great deal of research work on the mechanical properties of steel wire. In view of the important influence of heat treatment on the mechanical properties of steel wire, Wei et al. [6] studied the influence of low temperature annealing on the mechanical properties of cold-drawn pearlite stainless steel wire, and found that the tensile strength of steel wire was improved after low temperature annealing. Subsequently, Wei et al. [7] carried out an analysis of the effect of drawing heating on the microstructure and mechanical properties of cold-drawn pearlite stainless steel wire and found that low-temperature drawing gives lower strength and better ductility due to the lower content of nanocrystalline cementite. Beretta et al. [8] established a fatigue strength prediction model based on the propagation of surface defects in the cold drawing manufacturing process of steel wire. The model mainly considers the number of extreme defects in steel wire and the material properties jointly expressed by cyclic yield strength and crack propagation threshold. The manufacturing temperature of cold-drawn hypereutectoid steel wire is usually high. In order to explore the influence of high temperature on the evolution of its microstructure and mechanical properties, Jafari et al. [9] studied the changes of microstructure and tensile strength of steel wire after deformation at different temperatures for a certain period of time. They found that high temperature promoted the formation of carbon-poor (Fe, Mn, Cr) 3C cementite particles, thus destroying the stability of the layered structure and further reducing the strength of the steel wire. Nguyen et al. [10] explained the mechanical property degradation mechanism of steel wire under stress relaxation by using microstructure evolution and the grain boundary strengthening mechanism. Cruzado et al. [11,12] established a finite element model that can effectively predict the wear marks of steel wires under fretting wear conditions, which provides reference for wire rope designers to analyze the wear degree of wire ropes under different operating parameters and mechanism parameters.

The mechanical properties of wire rope directly determine the safety of mine hoisting. Some scholars have performed much research work on the mechanical properties of wire rope by means of experiments. Mouradi et al. [13] monitored the damage evolution process of 19 × 7 non-rotating wire ropes through fracture tensile tests, and defined the different stages of damage evolution of the wire rope and the critical life fraction that may lead to sudden failure. In view of the negative influence of broken wires on the mechanical properties of wire ropes, Zhang et al. [14] conducted bending fatigue tests on the wire rope samples with different pre-broken wire distributions based on a self-made bending fatigue test device and found that broken wires increased the stress in the internal strands and the contact force between steel wires, thus reducing the service life of wire ropes. The friction and wear of wire ropes seriously threatens the safety and reliability of traction transmission equipment. In order to truly simulate the friction and wear behavior of two wire ropes when they cross contact, Chang et al. [15,16,17,18,19,20,21,22] made a self-made wire rope friction and wear testing machine, and tests with different working conditions, structural parameters, and environmental parameters were carried out. In addition, Peng et al. [23] set up a testing machine to simulate the impact of winding hoisting wire ropes, and found through tests that the increase of load, slip speed, and impact speed seriously threatened the mechanical properties of the wire ropes.

Lubrication is an important measure to improve the service life and operational safety of wire ropes. However, the operating conditions of multi-layer winding hoisting wire ropes in an ultra-deep mine are often extremely harsh, and lubrication failure readily occurs, thus accelerating the wear of the wire ropes. In order to adapt to the harsh operating environment, modification of lubricating grease has been favored by many scholars. Zhao et al. [24] evaluated the friction and wear properties of lithium-based grease with nano-calcium borate as additive through an oscillating friction and wear tester. They found that the deposited nano-calcium borate and tribochemical compounds such as B_2_O_3_, CaO, and iron oxide on the friction surface were the main reasons for the improvement of the anti-wear and bearing capacity of grease. Bai et al. [25] used gallium-based liquid metal as an additive and uniformly added it to grease by means of mechanical stirring and ball milling and verified the improvement of the lubricating ability of the grease by four-ball tests. Graphene has a thin solid lubricating film, so it can effectively reduce friction and adhesion between contact surfaces, and can be used as an excellent anti-wear material [26]. Wang et al. [27] found that graphene as an additive could significantly improve the anti-wear and anti-wear ability of grease. The main reason is that graphene can not only be used as a protective substrate for deposited films but can also promote the formation of Fe_2_O_3_ and Li_2_O friction films, thus significantly improving the tribological properties of grease. Cheng et al. [28] prepared graphene-based semi-solid grease by a high dispersion mixing method and verified its good lubricating performance by friction tests. Sun et al. [29] verified that grease with graphene and nanographite as additives could effectively slow down the wear degree between steel wires with the help of a steel wire fretting wear testing machine. However, the chemical inertia of graphene and the mutual stacking of π–π bonds between layers make it difficult to uniformly disperse into lubricants, thus limiting its application. Ci et al. [30] produced fluorinated graphene which can be uniformly dispersed into base oil, while the lubricating oil modified by the additive has good wear-reducing characteristics. Due to the oxygen-containing functional groups on the substrate and the edge of the sheet, it is possible to modify graphene oxide to improve its dispersibility in lubricating oil. Fan et al. [31] prepared modified graphene oxide using alkyl imidazole ionic liquids as raw materials by the epoxy ring-opening reaction, cation-π stacking, or the van der Waals reaction. They found that the alkyl imidazole ionic liquid-graphene-rich friction film formed on the sliding surface was the main reason for its use as additive to improve the anti-friction and anti-wear properties of lubricating oils. Paul et al. [32] studied the effect of dodecylamine functionalized graphene as nano-additive on the tribological properties of industrial engine oil based on a UMT-2 friction testing machine and found that the friction film formed by the nano-additive reduced the coefficient of friction between friction pairs.

To sum up, previous scholars mainly studied the mechanical properties and friction and wear characteristics of steel wires and wire ropes. In addition, some scholars realized wear reduction by modifying lubricating grease. However, at present, there has been no research on modifying lubricating oil with lanthanum stearate to improve the anti-friction and anti-wear abilities of wire ropes under complex working conditions. Rare earth metal elements have strong chemical activity due to their special electronic layer structure and low electronegativity. They can diffuse and penetrate the subsurface layers of the friction contact surfaces to improve the structural properties of the materials and promote wear resistance; in addition, the corrosion resistance of the materials was also significantly enhanced. In this paper, first, lanthanum stearate was prepared by the saponification reaction, and its dispersion stability in IRIS was analyzed. Then, the extreme pressure performance, anti-friction, and anti-wear properties of LSMLO were investigated by four-ball friction tests. Finally, the influence of LSMLO on sliding friction and wear characteristics of wire ropes was analyzed by using a self-made wire rope sliding wear test rig.

## 2. Experimental Details

### 2.1. Preparation and Characterization of Lanthanum Stearate

Lanthanum chloride heptahydrate (LaCl_3_·7H_2_O, 99.7%), sodium hydroxide (NaOH, 98%) and stearic acid [CH_3_(CH_2_)_16_COOH, 98%] were all purchased from Shanghai McLean Biochemical Technology Co., Ltd. Lanthanum stearate was prepared by saponification. The reaction equations are shown in formulae (1) and (2). Lanthanum chloride heptahydrate, sodium hydroxide, and stearic acid were reacted in the molar mass ratio of 1:3:3. First, 1 g of lanthanum chloride heptahydrate was added into a conical flask containing 300 mL of deionized water and 100 mL of absolute ethanol (the purpose of adding absolute ethanol is to fully dissolve the stearic acid and make the reaction more efficient). After magnetic stirring for 10 min, 0.323 g of sodium hydroxide was added, and heating with magnetic stirring for 30 min carried out. With the addition of NaOH, the solution became turbid because lanthanum hydroxide precipitate was generated. Then the temperature was raised to 80 °C (DEG C), and 2.294 g of stearic acid was added, magnetic stirring was continued for 3 h. Finally, the precipitates were filtered, washed, and dried in vacuum, and the white solid powder of lanthanum stearate obtained.

(1)
$$LaCl_{3}\cdot 7H_{2}O+3NaOH\overset{\Delta }{\to }La{\left(OH\right)}_{3}↓+3NaCl+7H_{2}O,$$


(2)
$$La{\left(OH\right)}_{3}+3CH_{3}{\left(CH_{2}\right)}_{16}COOH\overset{80{}^{\circ}C,3h}{\to }{\left[CH_{3}{\left(CH_{2}\right)}_{16}COO\right]}_{3}La↓+3H_{2}O]$$


The chemical structure of lanthanum stearate was analyzed by a Thermo Fisher Nicolet IS10 infrared spectrometer (FT-IR), its thermal stability by a STA409PC thermal analyzer (TG), its crystal structure by a Bruker D8 Advance X-ray diffractometer (XRD), and its surface morphology by a Zeiss evo18 scanning electron microscope (SEM). Lanthanum stearate powders were added to IRIS according to the mass fraction ω = 0.05%, 0.1%, 0.2%, 0.5% (hereinafter abbreviated as wt.%), heated and stirred for 30 min, and then ultrasonically dispersed for 30 min to prepare uniformly dispersed modified lubricating oil.

### 2.2. Four-Ball Friction Test

The SGW-10A four-ball machine [29] (Figure 1 is the schematic diagram) was used to evaluate the extreme pressure, friction, and wear properties of the modified oil according to the test conditions in Table 1.

### 2.3. Sliding Wear Test of Wire Rope

The anti-wear characteristics of LSMLO on hoisting wire ropes were evaluated by using the wire rope sliding wear test rig [16], as shown in Figure 2. A 6 19+FC right intertwist hot galvanized hemp core wire rope was used in the test. This kind of wire rope was made of 6 independent rope strands (19 wires per strand) and a man-made fiber hemp core in the middle. The chemical composition of the wire rope is shown in Table 2, the structural parameters in Table 3 and the test conditions in Table 4. The wear area of the wire rope was obtained by post-processing software of the optical microscope.

## 3. Results and Discussion

### 3.1. Chemical and Structural Characterization of Lanthanum Stearate

Figure 3 shows the infrared absorption spectra of stearic acid and lanthanum stearate. The weak absorption peaks of stearic acid at the wavenumbers of 2955 cm^−1^ and 1104 cm^−1^ are attributed to the antisymmetric stretching vibration and plane rocking vibration (υ_as_ CH_3_ and ρ CH_3_) of methyl CH_3_ at one end of the octadecyl chain. The strong vibration peaks at the wavenumbers of 2917 cm^−1^, 2849 cm^−1^, 1472 cm^−1^, 1298 cm^−1^, and 719 cm^−1^ are attributed to the antisymmetric stretching vibration (υ_as_ CH_2_), symmetric stretching vibration (υ_s_ CH_2_), shear vibration (δ CH_2_), non-planar rocking vibration (ω CH_2_), and planar rocking vibration (ρ CH2) of the octadecyl chain methylene CH_2_, respectively [33]. Because one end of the stearic acid molecule is a carboxyl group (COOH), strong vibration peaks appear at the wavenumbers of 3456 cm^−1^, 1702 cm^−1^, and 941 cm^−1^, which are attributed to the stretching vibrations of the hydroxyl (υ OH), carbonyl group in the carboxyl group (υ C=O) and the non-plane swinging vibration of the hydroxyl group (ω OH), respectively. The stretching vibration peak (υ C=O) of the carbonyl group and the non-planar rocking vibration peak (ω OH) of the hydroxyl group belonging to stearic acid almost disappear in the infrared spectrum of lanthanum stearate. At the same time, the antisymmetric stretching vibration and symmetric stretching vibration peaks (υ_as_ C(O)O-, υ_s_ C(O)O-) of carboxylate ions appear at the wave numbers of 1528 cm^−1^ and 1408 cm^−1^ respectively, which is due to the fact that lanthanum interacts with carboxylate anions mainly through ionic bonds. It also indicates that most stearic acids have reacted to form lanthanum stearate.

Figure 4 presents the thermogravimetric map of lanthanum stearate. It is clear that lanthanum stearate has two obvious mass loss processes in the temperature range of 30–500 °C. The mass loss between 30–120 °C is about 1.73%, mainly from the thermal decomposition of residual water molecules, and the mass loss between 250–500 °C is about 48.76%, which is the main mass loss of lanthanum stearate and comes from the thermal decomposition of alkyl chain 35. After 500 °C, the quality loss is slow, mainly 4.69% between 650–740 °C. Thermogravimetric analysis shows that lanthanum stearate has good thermal stability and less decomposition before 250 °C, so lanthanum stearate, as a solid additive of lubricating oil, can theoretically be used for lubrication at higher temperature.

The XRD patterns of stearic acid and lanthanum stearate are presented in Figure 5. Stearic acid has characteristic diffraction peaks at diffraction angles 2θ = 4.4°, 6.6°, 8.82°, 11.04°, 15.48°, 21.5°, and 24.06°. However, the diffraction peaks at the above positions in the XRD pattern of lanthanum stearate disappear, and new characteristic diffraction peaks appear at the diffraction angles 2θ = 5.36°, 7.14°, 8.92°, 10.7°, 12.48°, 14.28°, 16.06°, 17.86°, and 19.66°. In addition, it is clear that the diffraction angle intervals between adjacent characteristic diffraction peaks are basically 1.78°. According to the Bragg equation, the crystal plane spacing corresponding to the above diffraction peaks can be calculated as 16.47 Å, 12.37 Å, 9.91 Å, 8.26 Å, 7.09 Å, 6.20 Å, 5.51 Å, 4.96 Å, 4.51 Å, respectively. It can be found that 16.47 is about 2 times of 8.26, about 3 times of 5.51, 12.37 is about 2 times of 6.20, and 9.91 is about 2 times of 4.96, that is, the crystal plane spacing is about three types of integer multiple relations, which indicates that lanthanum stearate is a disordered layered structure [33,34].

Figure 6 shows SEM images of lanthanum stearate. Lanthanum stearate is a typical lamellar structure with a sheet diameter of about several microns, a thickness of nanometer level, and irregular shape. Some small-sized fragments are randomly piled up on larger sheets or scattered everywhere. Some sheets are bonded together and form pore characteristics, which indicates that the prepared lanthanum stearate has many defects. Figure 7 presents EDS plane scan images of lanthanum stearate, and Figure 8 is an element content diagram of lanthanum stearate. The white-gray areas in Figure 7a are lanthanum stearate, the black areas are carbon conductive tape, so C element (Figure 7c) is densely distributed in the black areas, but not so densely distributed in the white-gray areas. However, La element (Figure 7e) and O element (Figure 7d) are densely distributed in white-gray areas, and La element is basically absent in the black areas, which also confirms the statement that the white-gray areas are lanthanum stearate. In addition, there is a certain amount of Cl element in lanthanum stearate (Figure 7f). Since the presence of Na element was not found (Figure 8), it is speculated that the residual Cl element is caused by the incomplete reaction of LaCl_3_·7H_2_O. Because the content of Cl element is extremely small, the atomic content is only 0.4%, so the influence here is negligible.

### 3.2. Dispersion Stability of Lanthanum Stearate in IRIS

Figure 9 presents the variation of the dispersion stability of lanthanum stearate with different concentrations in IRIS with time. It can be seen that the precipitation rate of lanthanum stearate becomes obviously faster with an increase in the addition amount. After standing for 2 h, a certain amount of precipitation was produced at the bottom of the oil sample with the addition amount of 0.5 wt.%, but when the addition amount was less than 0.2 wt.%, the precipitation was not obvious. After standing for 24 h, the thickness of the sediment layer at the bottom of the oil sample with added 0.5 wt.% of lanthanum stearate was relatively large. The color of the oil sample also changed from the original white yellowish brown to a darker yellow, and the delamination phenomenon was obvious. In addition, the bottom of the oil sample with added 0.2 wt.% lanthanum stearate also produced partial precipitation. However, the precipitations of oil samples with the addition amount of 0.05 wt.% and 0.1 wt.% were inconspicuous, indicating that the smaller the addition amount, the slower was the precipitation. After standing for 48 h, the precipitation under various amounts of addition was not much different from that after standing for 24 h, which indicates that the precipitation mainly occurs in the first day. Although there are octadecyl chains in its molecular structure, when the amount of lanthanum stearate is large, the octadecyl chains are aggregated by dispersion forces due to the collision of molecules in Brownian motion. However, when the addition amount is small, the molecules are less likely to collide in Brownian motion, so the probability of agglomeration is also reduced and the precipitation is slower.

In order to further quantitatively analyze the dispersion stability of lanthanum stearate at low concentration, the absorbance values of the upper oil samples were measured. Figure 10 presents the evolution of absorbance of an oil sample added with 0.1 wt.% lanthanum stearate over time. It is clear that lanthanum stearate has an obvious absorption peak in the wavelength range of 400–800 nm, which corresponds to visible light absorption. After standing for 1 day, 2 days, 5 days, and 10 days, the relative absorbance was 57.16%, 47.22%, 36.76%, and 22.07%, respectively. This may be due to the fact that the dispersion force provided between the alkyl chain of lanthanum stearate and the hydrocarbon group of IRIS is equivalent to its own weight. With the increase of aggregates, the dispersion force is less than its own weight, which eventually leads to its precipitation. In addition, with the increase of days standing, the absolute value of the slope of the relative absorbance polyline gradually decreases, which indicates that the precipitation rate of lanthanum stearate in IRIS gradually slows down with the extension of time.

In order to further improve the dispersion stability of lanthanum stearate in IRIS, that is, reduce its precipitation rate, this paper further investigated the effects of three dispersants (oleic acid, oleamine, span80) on the dispersion stability of lanthanum stearate in IRIS. In the tests, the addition amount of lanthanum stearate was 0.1 wt.%, and the addition amount of three kinds of dispersants was 2 wt.%. Before the tests, the mixtures were dispersed by heating, stirring and ultrasonic treatment for 20 min each to ensure uniform diffusion. Oil samples dispersed by oleic acid, oleamine, and span80 have obvious absorption peaks in the wavelength range of 400–800 nm, and oleic acid has the best dispersion effect, followed by oleamine while span80 is the worst, as shown in Figure 11a–c. In addition, the relative absorbance of lanthanum stearate is the highest under oleic acid, which indicates that its amount of precipitation is very small, and the dispersion stability of lanthanum stearate in IRIS has been greatly improved compared with the addition of the other two dispersants and no dispersant. Therefore, 2 wt.% oleic acid was added to ensure the dispersion stability of lanthanum stearate in IRIS in the following experimental studies on tribological characteristics.

### 3.3. Tribological Properties of Lanthanum Stearate Modified Oil

#### 3.3.1. Four-Ball Friction Test

Table 5 shows the last non-seizure load values (P_B_) of IRIS and LSMLOs adding 0.1 wt.%, 0.2 wt.% and 0.5 wt.% lanthanum stearate. The P_B_ value of LSMLO increases monotonically with the increase of addition amount. When the addition amount of lanthanum stearate is 0.1 wt.%, the P_B_ value of the modified oil increases by 13% compared with IRIS, and when the addition amount is 0.5 wt.%, the P_B_ value of the modified oil increases by 30%.

Figure 12 presents the variation curves of the coefficient of friction (CoF) with time of LSMLO with different additive amounts. Figure 12b and Figure 12c respectively present the curves of average CoF and average scar diameter of LSMLO with different additive amounts. The average CoF and the average wear scar diameter of IRIS are 0.146 mm and 0.730 mm, respectively. After adding 0.05 wt.% lanthanum stearate, the CoF curve of the modified oil basically coincides with IRIS, and the average CoF and average wear scar diameter are 0.145 mm and 0.677 mm, respectively, which are 0.3% and 7% less than IRIS. This indicates that when the addition amount is too small, the anti-friction ability of the modified oil is basically not improved, but the anti-wear ability is slightly improved. When the addition amount of lanthanum stearate exceeds 0.05 wt.%, the average CoF of modified oil presents an obvious trend of decreasing at first and then increasing, and the minimum value is reached when the addition amount is 0.2 wt.%. The average wear scar diameter of modified oil decreases monotonically with increasing addition amount. When the addition amount is 0.2 wt.%, the average CoF and average wear scar diameter of the modified oil are 0.095 mm and 0.551 mm, respectively, which are 35% and 25% lower than IRIS. However, when the addition amount is 0.5 wt.%, the average CoF and average wear scar diameter of the modified oil are 0.105 mm and 0.532 mm, respectively, which are higher than the addition amount of 0.2 wt.%, and the average wear scar diameter is slightly reduced. It is speculated that the explanation is that when the amount of lanthanum stearate is too much, the friction between lanthanum stearate molecules increases, which leads to the increase of CoF. However, due to its higher dispersion density and denser coating on the friction surface, the wear can still be further reduced. Considering that the dispersibility of lanthanum stearate in IRIS deteriorates sharply with the increase of its addition amount, the optimal addition amount is 0.2 wt.%.

Figure 13a and Figure 13b respectively show the SEM images of wear scars lubricated by IRIS and LSMLO, in which the addition amount of lanthanum stearate is 0.5 wt.%. Under IRIS lubrication, the furrows on the wear scar surface are densely distributed but the depth is shallow, and part of the wear debris adheres to the surface of the wear scar. It can be seen that the wear mechanism is mainly abrasive wear, accompanied by a small amount of adhesive wear. After lubrication with LSMLO, the wear scar size is obviously smaller at the same magnification, which indicates that the wear amount is significantly reduced. However, the width and depth of the furrows on the worn surface are increased, and they are rougher and uneven than those in IRIS. Therefore, the abrasive wear is relatively serious, while the adhesive wear is not obvious. This may be related to the agglomeration of LSMLO.

#### 3.3.2. Wire Rope Sliding Wear Test

Figure 14 presents the comparisons of CoF and wear area between IRIS and LSMLO under different contact loads. The average value of CoF at the last 1200 mm of the sliding distance was taken as the average CoF. In the case of IRIS lubrication, the CoF under each contact load showed a trend of first increasing, then decreasing and finally stabilizing with an increase in sliding distance (Figure 14a). When the contact load is 230 N, the CoF presents a zigzag shape in the first and middle period, which may be related to the destruction of boundary layer caused by high temperature due to higher load. Then, under the repair of the oil film, the CoF gradually decreases and reaches a stable level. Under LSMLO lubrication, the overall CoF curves are relatively stable, but they tend to increase when the contact loads are large, as shown in Figure 14b. The average CoFs under IRIS and LSMLO lubrication have trends of first increasing and then decreasing with increasing the contact load (Figure 14c). However, the friction characteristics of LSMLO are greatly affected by the contact loads. The effect of friction reduction is remarkable under low load, while it has the effect of increasing friction under high load. The wear areas of wire ropes increase with increasing contact load, but the wear areas under LSMLO lubrication decrease compared with IRIS (Figure 14d). When the contact load is 50 N, the wear area decreases by up to 84%. Therefore, compared with IRIS, LSMLO can significantly reduce the friction and wear under low load, but increase the friction, and the wear reduction is not obvious under high load.

Figure 15 shows the comparisons of CoF and wear area between IRIS and LSMLO at different sliding speeds. The average CoF between IRIS and LSMLO lubricated ropes decreases with increasing sliding speed, as shown in Figure 15c. When the sliding speed is 6 mm/s, the average CoF increases by 7% compared with IRIS lubrication, while when the sliding speed is 12 mm/s and 18 mm/s, the CoFs decrease by 1% and 40% compared with IRIS lubrication, respectively. The results indicate that LSMO increases friction at low sliding speed, while the friction reduction effect is obvious at high sliding speed. Under the lubrications of IRIS and LSMLO, the wear areas decrease with increasing sliding speed (Figure 15d). Compared with IRIS, the wear areas of LSMLO lubrication decrease by 29%, 18%, and 35% respectively at sliding speeds of 6 mm/s, 12 mm/s, and 18 mm/s. Therefore, LSMLO can reduce the wear between ropes at different sliding speeds, increase the friction at low sliding speed, and reduce the friction and have better anti-wear effect at high sliding speed than IRIS.

Figure 16 and Figure 17 present SEM images of wear scars of wire ropes lubricated with IRIS and LSMLO under different contact loads and sliding speeds (the wire with the largest wear scar area is selected for analysis). Under IRIS lubrication, when the test condition is 50 N+18 mm/s, there are furrows on the worn surface of the wire rope, and the phenomena of material spalling and scratching are obvious, and also accompanied by adhesion phenomenon. The wear mechanisms of wire ropes are mainly abrasive wear and adhesive wear. With the increase of contact load to 160 N, the furrow phenomenon is no longer obvious, the wear surface becomes smooth, and the spalling areas are obviously reduced, which indicates that the degree of abrasive wear and adhesive wear is reduced. However, when the contact load increases to 230 N, the micro-pit areas on the wear surface increase, and more wear debris are found, which indicates that abrasive wear and adhesive wear are more serious under this load. When the test condition is 100 N+6 mm/s, the furrow phenomenon is obvious. With the increase of sliding speed, the furrow basically disappears, indicating that abrasive wear is more obvious when the sliding speed is lower. Under LSMLO lubrication, the wear surface is smoother than that under IRIS lubrication under the same test conditions, and the abrasive wear and adhesive wear are reduced. Abrasive wear and adhesive wear are also more obvious at low sliding speed (100 N + 6 mm/s), which is consistent with the higher friction and wear between ropes at low sliding speed.

## 4. Conclusions

In this study, first, lanthanum stearate was prepared, and its chemical and structural properties were characterized. Then, the dispersion stability of lanthanum stearate in IRIS was investigated. Finally, the tribological properties of LSMLO were analyzed. The main conclusions are as follows:Lanthanum stearate was successfully prepared by the saponification reaction. The prepared lanthanum stearate has a disordered layered structure with an obvious pore structure. The dispersion stability of lanthanum stearate in IRIS is obviously improved by oleic acid dispersion;With the increasing addition of lanthanum stearate, the P_B_ value of the modified oil increases monotonically, and the CoF first increases and then decreases. When the addition amount is 0.2 wt.%, the friction coefficient decreases by up to 35%, and the wear spot diameter decreases by 25%;Compared with IRIS, LSMLO can reduce the wear between ropes at various sliding speeds and contact loads, but it has better anti-wear and antifriction performance at high speed and low load.

## Figures and Tables

**Figure 1 materials-14-05821-f001:**
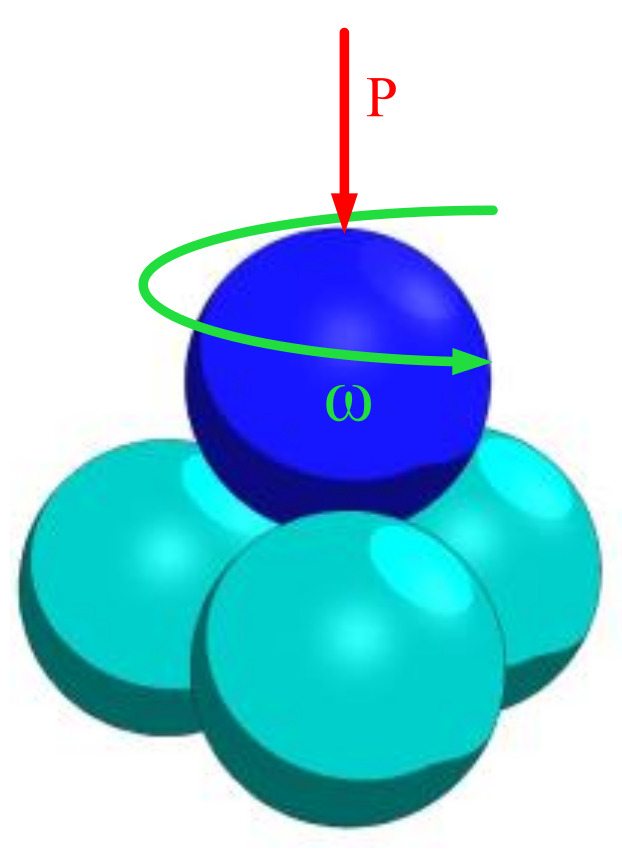
Schematic diagram of four-ball machine.

**Figure 2 materials-14-05821-f002:**
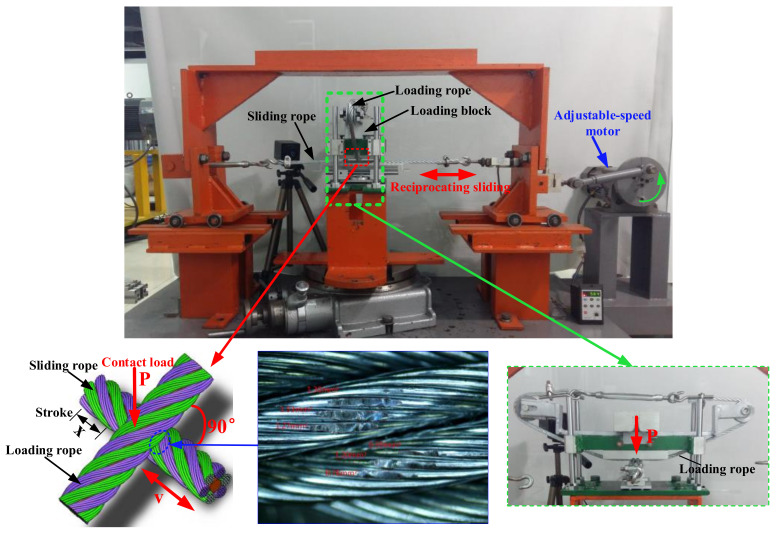
Wire rope sliding wear test rig.

**Figure 3 materials-14-05821-f003:**
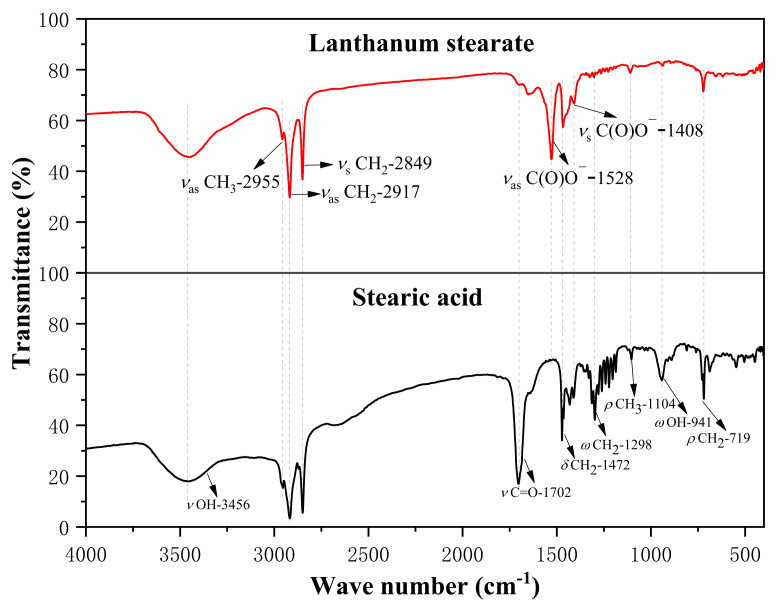
Infrared spectra of stearic acid and lanthanum stearate.

**Figure 4 materials-14-05821-f004:**
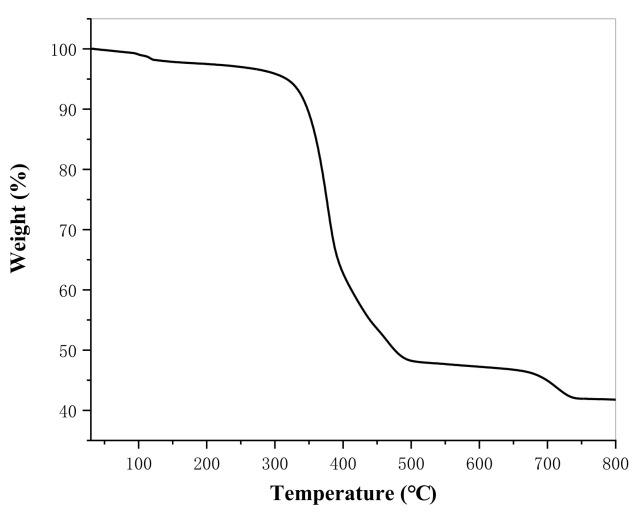
Thermogravimetric map of lanthanum stearate.

**Figure 5 materials-14-05821-f005:**
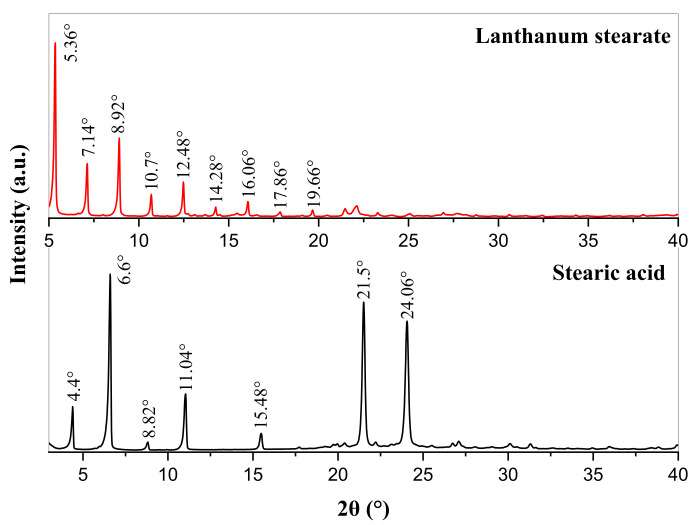
XRD patterns of stearic acid and lanthanum stearate.

**Figure 6 materials-14-05821-f006:**
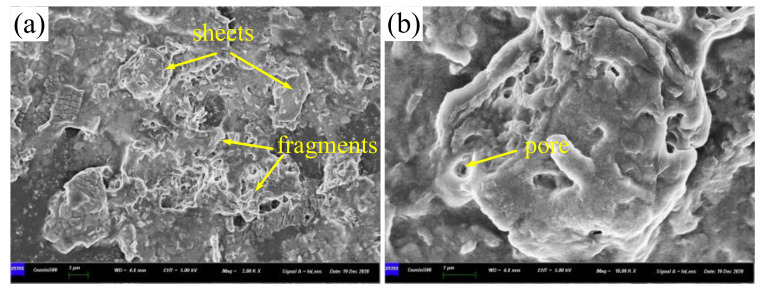
SEM images of lanthanum stearate. (**a**) 2.0 × 10^3^ times; (**b**) 1.0 × 10^4^ times.

**Figure 7 materials-14-05821-f007:**
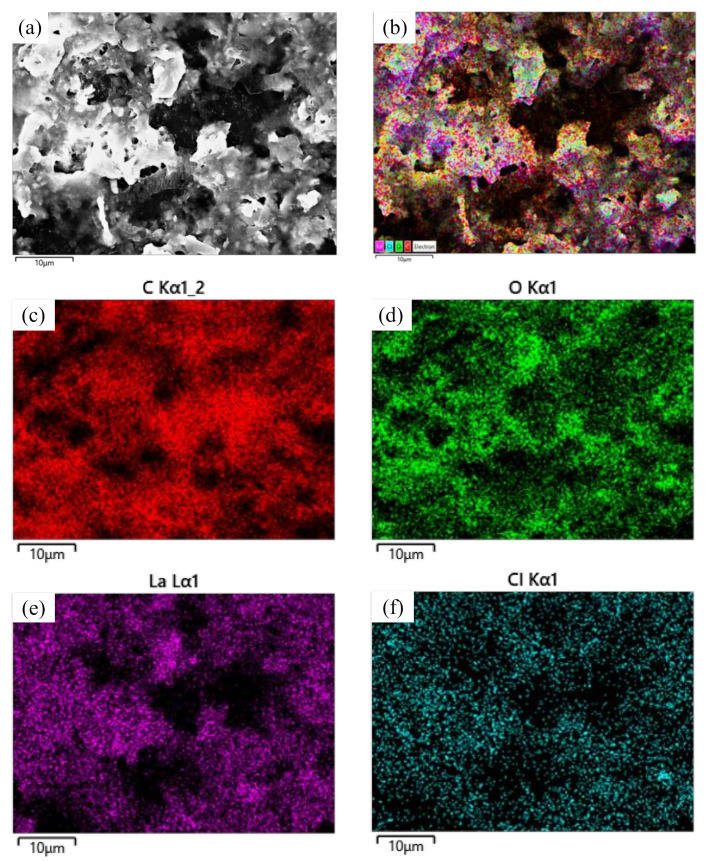
EDS plane scan images of lanthanum stearate. (**a**) lanthanum stearate; (**b**) Element distribution; (**c**) C; (**d**) O; (**e**) La; (**f**) Cl.

**Figure 8 materials-14-05821-f008:**
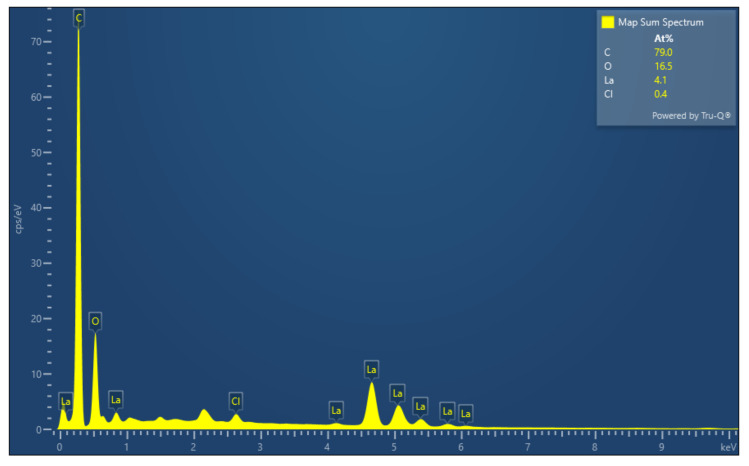
Element content diagram of lanthanum stearate.

**Figure 9 materials-14-05821-f009:**
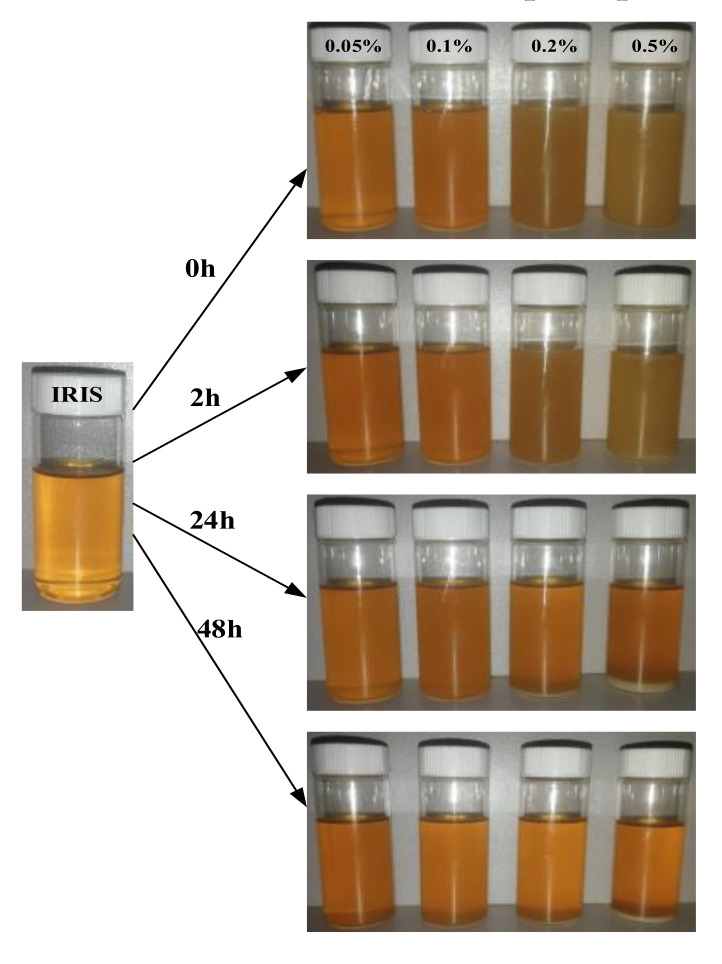
Dispersion stability of lanthanum stearate with different concentrations in IRIS with time.

**Figure 10 materials-14-05821-f010:**
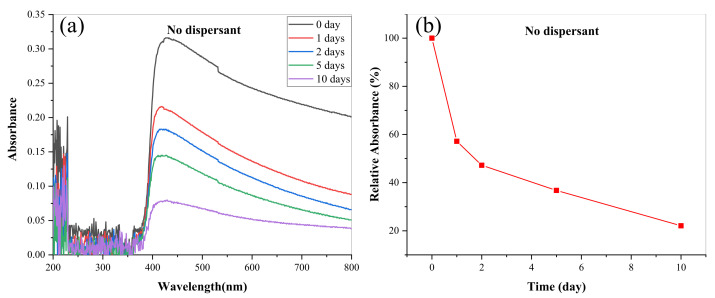
Evolution of absorbance of LSMLO with time. (**a**) change curves; (**b**) relative absorbance.

**Figure 11 materials-14-05821-f011:**
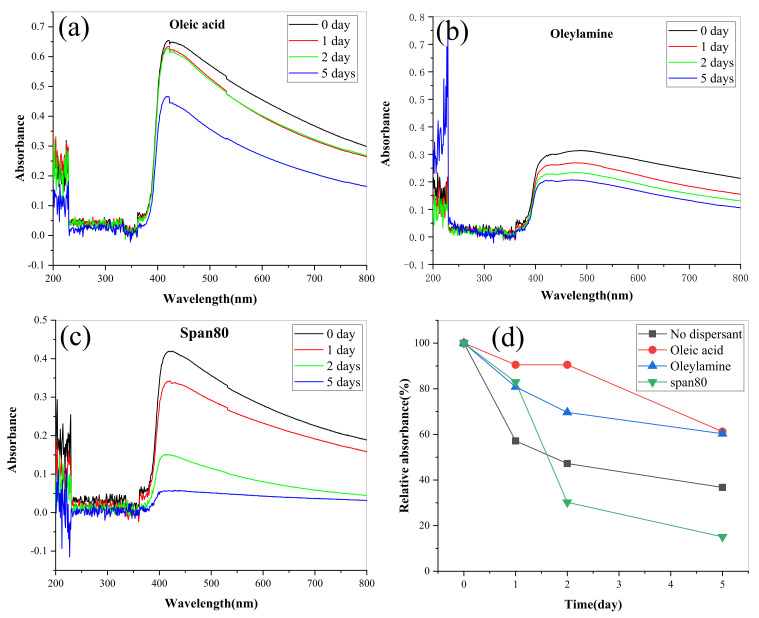
Variations of absorbance of LSMLO with different dispersants over time. (**a**) oleic acid; (**b**) Oleylamine; (**c**) span 80; (**d**) relative absorbance.

**Figure 12 materials-14-05821-f012:**
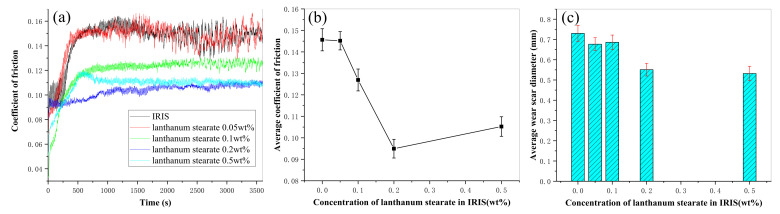
Variations of CoF and wear mark size of LSMLO with different addition amounts. (**a**) CoF; (**b**) average CoF; (**c**) average wear scar diameter.

**Figure 13 materials-14-05821-f013:**
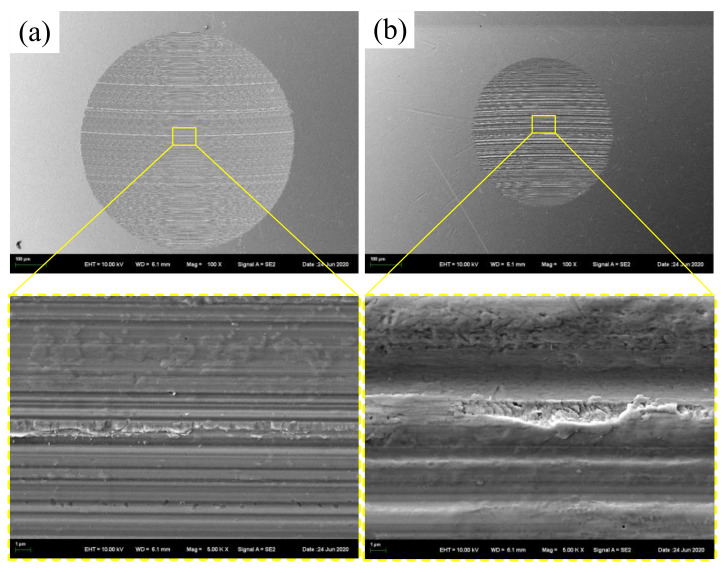
SEM images of wear scars after lubrication with IRIS and LSMLO. (**a**) IRIS; (**b**) LSMLO.

**Figure 14 materials-14-05821-f014:**
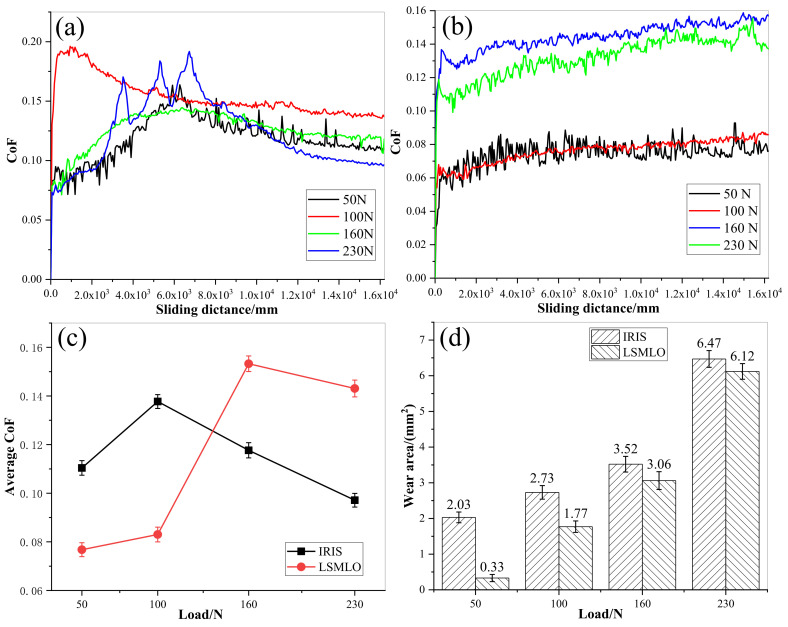
CoFs and wear areas of wire ropes under different contact loads lubricated with IRIS and LSMLO. (**a**) evolutions of CoF under IRIS lubrication; (**b**) evolutions of CoF under LSMLO lubrication; (**c**) comparison of average CoF between IRIS and LSMLO lubrication; (**d**) evolutions of wear area under IRIS and LSMLO lubrications.

**Figure 15 materials-14-05821-f015:**
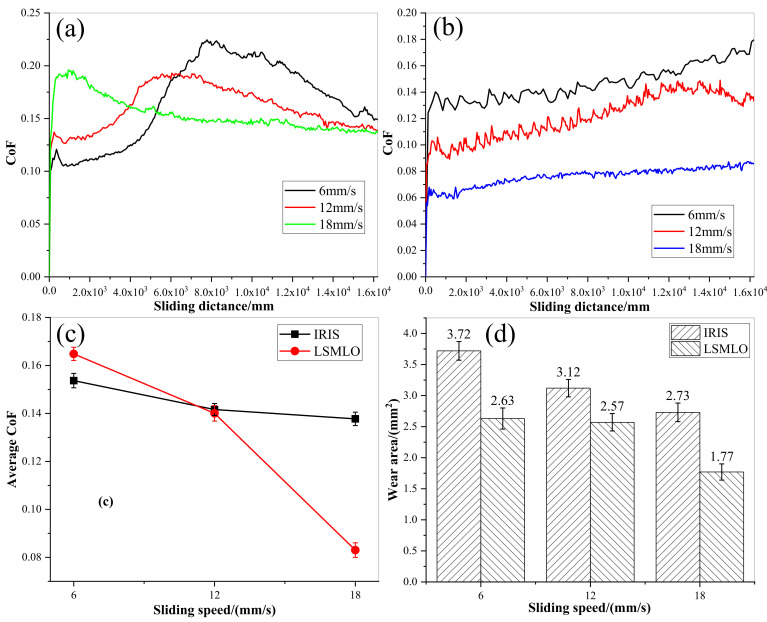
CoFs and wear areas of wire ropes under different sliding speeds lubricated with IRIS and LSMLO. (**a**) evolutions of CoF under IRIS lubrication; (**b**) evolutions of CoF under LSMLO lubrication; (**c**) comparison of average CoF between IRIS and LSMLO lubrication; (**d**) evolutions of wear area under IRIS and LSMLO lubrications.

**Figure 16 materials-14-05821-f016:**
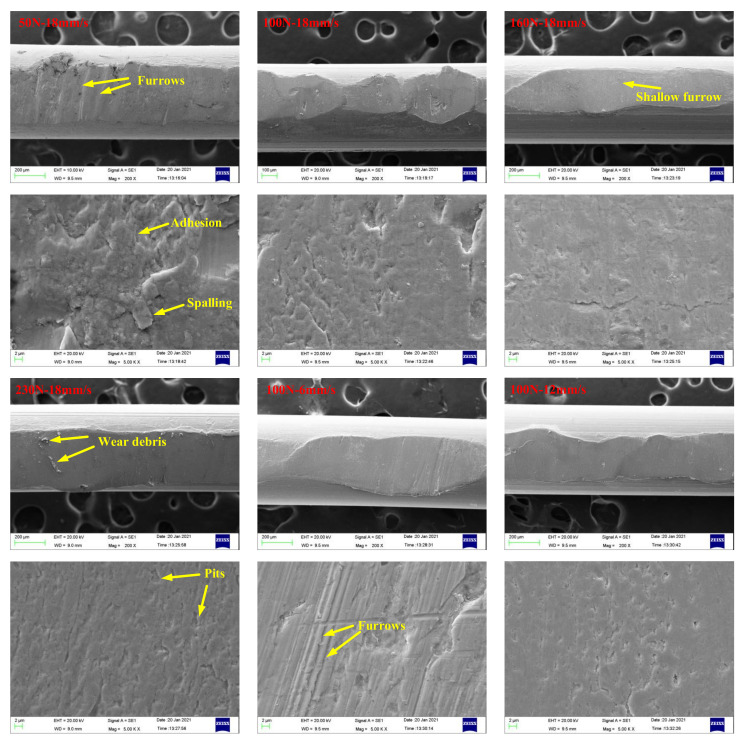
SEM images of wear scars of wire ropes lubricated with IRIS.

**Figure 17 materials-14-05821-f017:**
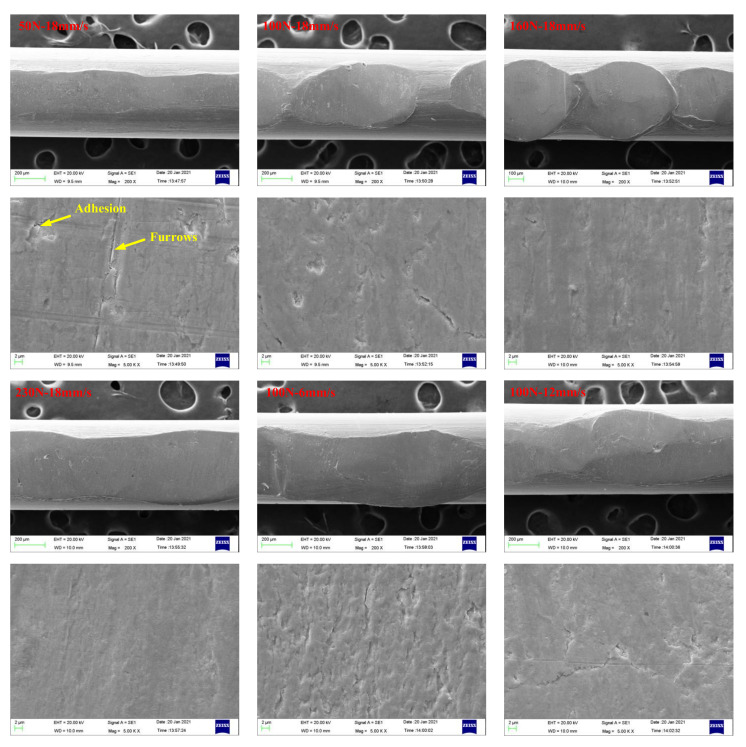
SEM images of wear scars of wire ropes lubricated with LSMLO.

**Table 1 materials-14-05821-t001:** Parameters of four-ball friction tests.

Test Parameters	Load (N)	Rotational Speed (r/min)	Time (s)	Temperature (°C)
Extreme pressure test	/	1450	10	25
Friction and wear test	147	1200	3600	50

**Table 2 materials-14-05821-t002:** Chemical element composition of the wire rope.

Element	Fe	Zn	C	Mn	Si	Ni	S	P
Content/%	94	4.53	0.87	0.39	0.02	0.01	<0.01	<0.01

**Table 3 materials-14-05821-t003:** Parameters of the wire rope.

Parameters	Diameter of Wire Rope/mm	Diameter of Wire/mm	Twist Angle of Strand/°	Strand Pitch/mm	Tensile Strength /MPa	Breaking Force/N
Value	9.3	0.6	15.5	70	1570	52,500

**Table 4 materials-14-05821-t004:** Sliding wear test conditions of wire ropes.

Test Parameters	Group 1	Group 2
Contact load/N	50/100/160/230	100
Sliding speed/(mm/s)	18	6/12/18
Crossing angle/°	90	90
Stroke/mm	20	20
Sliding distance/mm	16,200	16,200
Temperature	Room temperature	Room temperature
Relative humidity/%	60 ± 5	60 ± 5
Atmosphere	Laboratory air	Laboratory air

**Table 5 materials-14-05821-t005:** P_B_ values of IRIS and LSMLOs.

Oil	IRIS	IRIS+0.1 wt.% Lanthanum Stearate	IRIS+0.2 wt.% Lanthanum Stearate	IRIS+0.5 wt.% Lanthanum Stearate
PB	265 N	300 N	320 N	345 N

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
