# Peer review of "Preparation and Tribological Properties of Lanthanum Stearate Modified Lubricating Oil for Wire Rope in a Mine Hoist"

_materials, 2021, doi:10.3390/ma14195821_

Round 1
Reviewer 1 Report
The paper gives a study on the tribological characteristics of lanthanum stearate modified lubricating oil (LSMLO) was carried out. It is written in a very clear way with many given results of the conducted experiments. I have only one comment regarding the test parameters given in the Table 4. How they are selected, how the range for a contact load and sliding speed has been determined? Is there any design of experiment prepared? Please correct the PROE to PORE at the Fig. 6b).
Author Response
Dear Reviewer 1:
Thank you for the comments concerning our manuscript entitled “Tribological Properties of Lanthanum Stearate Modified Lubricating Oil for Wire Rope in Mine Hoist” (No.: materials-1368121). Those comments are all valuable and very helpful for revising and improving our paper, as well as the important guiding significance to our research. We have studied comments carefully and have made correction which we hope meet with approval. Revised portion are marked in red in the revised manuscript. The main corrections in the paper and the responds to the reviewer’s comments are as follows:’’
- I have only one comment regarding the test parameters given in the Table 4. How they are selected, how the range for a contact load and sliding speed has been determined? Is there any design of experiment prepared?
Reply: We designed the test parameters of the manuscript according to the actual working conditions of a mine hoist, so as to investigate tribological properties of lanthanum stearate modified lubricating oil for wire rope. Those test parameters are also described in the following literatures:
- Peng Y.X.; Chang X.D.; Zhu Z.C. Sliding friction and wear behavior of winding hoisting rope in ultra-deep coal mine under different conditions. Wear 2016, 368-369, 423-434.
- Chang X.D.; Peng Y.X.; Zhu Z.C. Effects of Strand Lay Direction and Crossing Angle on Tribological Behavior of Winding Hoist Rope. Materials 2017, 10(6), 1-20.
- Chang X.D.; Peng Y.X.; Zhu Z.C. Experimental investigation of mechanical response and fracture failure behavior of wire rope with different given surface wear. Int. 2018, 119, 208-221.
- Chang X.D.; Peng Y.X.; Zhu Z.C. Evolution properties of tribological parameters for steel wire rope under sliding contact conditions. Metals 2018, 8(10), 1-16.
- Chang X.D.; Peng Y.X.; Zhu Z.C. Effect of wear scar characteristics on the bearing capacity and fracture failure behavior of winding hoist wire rope. Int. 2019, 130, 270-283.
- Chang X.D.; Huang H.B.; Peng Y.X. Friction, wear and residual strength properties of steel wire rope with different corrosion types. Wear 2020, 458-459, 203425.
- Please correct the PROE to PORE at the Fig. 6b).
Reply: We are very sorry for this low-level mistake, and we have changed the PROE in Fig. 6(b) to PORE.
Finally, we appreciate for reviewer’s warm work earnestly, and hope that the corrections above will meet with approval.
Once again, thank you very much for your comments and suggestions.
Yours sincerely,
Qing Zhang

Reviewer 2 Report
The article, titled Tribological Properties of Lanthanum Stearate Modified Lubricating Oil for Wire Rope in Mine Hoist, has a very interesting topic and is written in a coherent and clear manner. The article structure and language are correct. It was nice to read such a well-written article.
This paper is very interesting for mining industry, it discus tribological properties of Lanthanum Stearate Modified Lubricating Oil. The aim of this work was to improve anti-friction and anti-wear abilities by modification of lubricating oil with lanthanum stearate, and it is supported by the results and conclusions. The text is well written and easy to fallow. Conclusions are consistent with presented results.
Author Response
Dear Reviewer 2:
We thank you very much for your affirmation of the research content of our manuscript entitled “Tribological Properties of Lanthanum Stearate Modified Lubricating Oil for Wire Rope in Mine Hoist” (No.: materials-1368121). Moreover, we appreciate for reviewer’s warm work earnestly, and send sincere blessings to the reviewer.
Yours sincerely,
Qing Zhang

Reviewer 3 Report
- If you make a statement that you obtained lanthanum stearate for the first time by saponification, it would be good to discuss this issue in more detail and give a description of other methods with corresponding references.
- Eq. 1 - LaCl3
- Could you explain, why was 90 degrees angle between loading and sliding wire ropes selected? Seems that in real application contacting sites of the wire ropes are parallel to each other.
- Line 182 - no need to place official standard to references
- Table 1 - the load value is absent for extreme pressure test - why?
- Table 2 - a standard for the wire rope material and manufacturer could be mentioned instead of enumeration of chemical elements. From the other side, there is only a mention about galvanization of the wire rope, but no data about the the coating (thickness, microhardness)
- No references regarding the IR and XRD spectra.
- Line 385 - In order to produce abrasive wear, lanthanum stearate aggregates should be harder then the cable material. Can you prove that it is really so?
- Fig. 10, 11 - it would be better to take a single scale for Y axes in all diagrams.
- Fig. 6, 7, 13, 16, 17 - no scale bars are shown
- Fig. 16, 17 - no explanations are given. Features, mentioned in the text, like furrows or micro-pit areas should be shown directly on the images.
Author Response
Dear Reviewer 3:
Thank you for the comments concerning our manuscript entitled “Tribological Properties of Lanthanum Stearate Modified Lubricating Oil for Wire Rope in Mine Hoist” (No.: materials-1368121). Those comments are all valuable and very helpful for revising and improving our paper, as well as the important guiding significance to our research. We have studied comments carefully and have made correction which we hope meet with approval. Revised portion are marked in red in the revised manuscript. The main corrections in the paper and the responds to the reviewer’s comments are as follows:’’
- If you make a statement that you obtained lanthanum stearate for the first time by saponification, it would be good to discuss this issue in more detail and give a description of other methods with corresponding references.
Reply: Thank you very much for the reviewer's question. Saponification is a common method of chemical reaction between hydroxyl and alkoxy. Saponification is used in this manuscript to effectively obtain lanthanum stearate, so it has not been discussed too much.
- Eq. 1 - LaCl3
Reply: We are very sorry for this low-level mistake, and we have changed the Lacl3 in Eq. 1 to LaCl3.
- Could you explain, why was 90 degrees angle between loading and sliding wire ropes selected? Seems that in real application contacting sites of the wire ropes are parallel to each other.
Reply: In this study, the contact form of winding hoisting wire rope under the condition of random rope was used to test the anti-wear performance of LSMLO, which is reflected in the following references:
- Peng Y.X.; Chang X.D.; Zhu Z.C. Sliding friction and wear behavior of winding hoisting rope in ultra-deep coal mine under different conditions. Wear 2016, 368-369, 423-434.
- Chang X.D.; Peng Y.X.; Zhu Z.C. Effects of Strand Lay Direction and Crossing Angle on Tribological Behavior of Winding Hoist Rope. Materials 2017, 10(6), 1-20.
- Chang X.D.; Peng Y.X.; Zhu Z.C. Experimental investigation of mechanical response and fracture failure behavior of wire rope with different given surface wear. Int. 2018, 119, 208-221.
- Chang X.D.; Peng Y.X.; Zhu Z.C. Evolution properties of tribological parameters for steel wire rope under sliding contact conditions. Metals 2018, 8(10), 1-16.
- Chang X.D.; Peng Y.X.; Zhu Z.C. Effect of wear scar characteristics on the bearing capacity and fracture failure behavior of winding hoist wire rope. Int. 2019, 130, 270-283.
- Chang X.D.; Huang H.B.; Peng Y.X. Friction, wear and residual strength properties of steel wire rope with different corrosion types. Wear 2020, 458-459, 203425.
- Line 182 - no need to place official standard to references
Reply: Thank you very much for the reviewer's suggestion. We have removed the reference standards.
- Table 1 - the load value is absent for extreme pressure test - why?
Reply: In the extreme pressure test, the load value, namely the last non-seizure load value (PB), was obtained by setting the rotating speed, running time and temperature. Therefore, the load value does not appear in Table 1.
- Table 2 - a standard for the wire rope material and manufacturer could be mentioned instead of enumeration of chemical elements. From the other side, there is only a mention about galvanization of the wire rope, but no data about the the coating (thickness, microhardness)
Reply: We are very sorry that it is difficult for us to obtain the thickness and hardness of galvanized layer of this kind of wire rope. This kind of steel wire rope has been tested in the literatures listed in the reply to the Question 3. Through the content of Zn in the element, we can know that the wire rope is slightly galvanized. Through the breaking force and tensile strength of the wire rope, we can obtain the mechanical properties of the wire rope.
- No references regarding the IR and XRD spectra.
Reply: Thank you very much for your comment. We have added references to the manuscript.
- Line 385 - In order to produce abrasive wear, lanthanum stearate aggregates should be harder then the cable material. Can you prove that it is really so?
Reply: We are very sorry that we can't prove this is true. We have made the following change for the reason:
Therefore, the abrasive wear is relatively serious, while the adhesive wear is not obvious. This may be related to the agglomeration of LSMLO.
- Fig. 10, 11 - it would be better to take a single scale for Y axes in all diagrams.
Reply: Thank you very much for the reviewer's suggestion. In Figs. 10 (a) and 11 (a-c), in order to better show the variation difference between curves, the scale of Y coordinate is indeed different, but in Figs. 10 (b) and 11 (d), the relative absorbance is compared with the unified Y coordinate, which we think can be clearly explained and hope the reviewer is satisfied.
- Fig. 6, 7, 13, 16, 17 - no scale bars are shown
Reply: There are corresponding scales in the lower left corner of Figs 6, 7, 13, 16 and 17.
- Fig. 16, 17 - no explanations are given. Features, mentioned in the text, like furrows or micro-pit areas should be shown directly on the images.
Reply: The reviewer 's suggestion is very good, and we have annotated it accordingly in Figs 16 and 17.
Finally, we appreciate for reviewer’s warm work earnestly, and hope that the corrections above will meet with approval.
Once again, thank you very much for your comments and suggestions.
Yours sincerely,
Qing Zhang

Reviewer 4 Report
The paper addresses an interesting subject with practical application: improving the lubricating conditions of wire ropes. The authors propose a new additive for the lubricating oil and test it on a standard 4-ball test rig and on an in-house made test-rig, specially designed for wire ropes.
The use of Lanthanum Stearate as an additive gives the originality of the work reported herein.
The paper has two main parts: firstly, the additive and the lubricant are thoroughly analysed from a chemical point of view. The second part of the paper is dedicated to the evaluation of the friction and anti-wear properties of the lubricant, additivated with Lanthanum Stearate in 4 concentrations.
In regard to this, I would say that the title does not reflect entirely the content, and the first part of the paper should be referred in the title.
My competence covers mainly the tribological studies presented in this manuscript, hence I will refer hereafter only to this part.
Broadly, this part is well organized, and the experiments performed correspondingly. However, some details should be clarified.
4-ball tests
1) The 4-ball tests were made according to 2 Chinese standards which are not available; probably they are very similar with well-known ASTM standards (ASTM D2783, ASTM D4172), world-wide accepted. I would recommend introducing references for these standards.
2) What is the temperature value represented in Table 1 ? Is this temperature constant during experiments? (most likely, not).
3) The extreme pressure test is done with continuously increasing loads; which are the values of these loads? What was the seizure load?
4)The values of the COF for IRIS (lubricant without additive) are greater than typical values obtained on similar tests; can the authors comment the obtained result, or eventually, compare it with some other results found in literature ?
Wire-rope sliding friction tests
Undoubtedly, these tests seem to be more realistic. However, the test rig and the procedure lack a clear description. It is, at least, necessary to use citations referring to previously published papers of the authors.
1) The tests reveal the influence of the relative sliding speed, and normal load. How is the "wear area" shown in Fig. 14.d obtained?
2) The wear rate was evaluated by the amount of debris collected; some details should be given on the procedure.
3) At page 13 (lines 392-410, and not only) there is a detailed description of what we clearly see on the graphs shown in Fig. 14. Probably a shortened text will make the paper easier to read.
4)The authors are kindly asked to argue the text written at lines 397-399:
"...indicating that the oil film breaks and temporary local dry friction occurs between wire ropes. Then, under the repair of the 398 oil film, the CoF gradually decreases and reaches a stable level."
In my opinion some chemical reactions produced by the temperature increase could be at the origin of "boundary layer destruction". It is difficult to accept the idea of "dry friction".
5) A comparison of these results with other obtained by the authors on the same test-rig seems to be necessary.
Author Response
Dear Reviewer 4:
Thank you for the comments concerning our manuscript entitled “Tribological Properties of Lanthanum Stearate Modified Lubricating Oil for Wire Rope in Mine Hoist” (No.: materials-1368121). Those comments are all valuable and very helpful for revising and improving our paper, as well as the important guiding significance to our research. We have studied comments carefully and have made correction which we hope meet with approval. Revised portion are marked in red in the revised manuscript. The main corrections in the paper and the responds to the reviewer’s comments are as follows:’’
- In regard to this, I would say that the title does not reflect entirely the content, and the first part of the paper should be referred in the title
Reply: The reviewer 's question is very good, and we have adjusted the title of the paper accordingly: Preparation and Tribological Properties of Lanthanum Stearate Modified Lubricating Oil for Wire Rope in Mine Hoist.
4-ball tests
- The 4-ball tests were made according to 2 Chinese standards which are not available; probably they are very similar with well-known ASTM standards (ASTM D2783, ASTM D4172), world-wide accepted. I would recommend introducing references for these standards.
Reply: Table 1 lists the specific working condition parameters of these two standards, so we have removed the reference of these two standards.
- What is the temperature value represented in Table 1? Is this temperature constant during experiments? (most likely, not).
Reply: The temperatures listed in Table 1 are the set temperatures of the four-ball machine, that is, the temperatures are constant at the set values in the operating space of the balls.
- The extreme pressure test is done with continuously increasing loads; which are the values of these loads? What was the seizure load?
Reply: In the extreme pressure test, the load value, namely the last non-seizure load value (PB), was obtained by setting the rotating speed, running time and temperature. Therefore, the load value does not appear in Table 1.
- The values of the COF for IRIS (lubricant without additive) are greater than typical values obtained on similar tests; can the authors comment the obtained result, or eventually, compare it with some other results found in literature?
Reply: Thank you very much for the reviewer 's question. The CoF of IRIS is larger than that of LSMLO, and the reason for this phenomenon has been explained in this paper: Lamellar lanthanum stearate additive improves the anti-wear characteristics of IRIS.
Wire-rope sliding friction tests
Undoubtedly, these tests seem to be more realistic. However, the test rig and the procedure lack a clear description. It is, at least, necessary to use citations referring to previously published papers of the authors.
Reply: The test rig in the following references are described in detail, so this paper only carried out a simple introduction, and the literatures are cited.
- Peng Y.X.; Chang X.D.; Zhu Z.C. Sliding friction and wear behavior of winding hoisting rope in ultra-deep coal mine under different conditions. Wear 2016, 368-369, 423-434.
- Chang X.D.; Peng Y.X.; Zhu Z.C. Effects of Strand Lay Direction and Crossing Angle on Tribological Behavior of Winding Hoist Rope. Materials 2017, 10(6), 1-20.
- Chang X.D.; Peng Y.X.; Zhu Z.C. Experimental investigation of mechanical response and fracture failure behavior of wire rope with different given surface wear. Int. 2018, 119, 208-221.
- Chang X.D.; Peng Y.X.; Zhu Z.C. Evolution properties of tribological parameters for steel wire rope under sliding contact conditions. Metals 2018, 8(10), 1-16.
- Chang X.D.; Peng Y.X.; Zhu Z.C. Effect of wear scar characteristics on the bearing capacity and fracture failure behavior of winding hoist wire rope. Int. 2019, 130, 270-283.
- Chang X.D.; Huang H.B.; Peng Y.X. Friction, wear and residual strength properties of steel wire rope with different corrosion types. Wear 2020, 458-459, 203425.
- The tests reveal the influence of the relative sliding speed, and normal load. How is the "wear area" shown in Fig. 14.d obtained?
Reply: Thank you very much for the reviewer 's question. The wear area of the wire rope is obtained by post-processing software of optical microscope. We have added it in the details of the test.
- The wear rate was evaluated by the amount of debris collected; some details should be given on the procedure.
Reply: We are very sorry that we only analyze the micro-morphology of the worn surfaces of the wire ropes, but do not analyze the wear rate by debris.
- At page 13 (lines 392-410, and not only) there is a detailed description of what we clearly see on the graphs shown in Fig. 14. Probably a shortened text will make the paper easier to read.
Reply: Our writing idea is to first describe the phenomenon in the figure in detail, and then summarize it in a short sentence. For example, after describing Fig. 14 in detail, we summarize it in the following words: Therefore, compared with IRIS, LSMLO can significantly reduce the friction and wear under low load, but increase the friction, and the wear reduction is not obvious under high load.
- The authors are kindly asked to argue the text written at lines 397-399:
"...indicating that the oil film breaks and temporary local dry friction occurs between wire ropes. Then, under the repair of the 398 oil film, the CoF gradually decreases and reaches a stable level."
In my opinion some chemical reactions produced by the temperature increase could be at the origin of "boundary layer destruction". It is difficult to accept the idea of "dry friction".
Reply: The question raised by the reviewer have benefited us a lot, and we have made the following modifications to the contents of this part:
When the contact load is 230 N, the CoF presents a zigzag shape in the first and middle period, which may be related to the destruction of boundary layer caused by high temperature caused by higher load.
- A comparison of these results with other obtained by the authors on the same test-rig seems to be necessary.
Reply: Several literatures have been published through this test-rig, and its repeatability can stand the test. The test conditions in this paper are different from the previous literatures. We think it is difficult to compare with the previous test results, but we have carried out repeated tests to verify the effectiveness of the results, and there is no conflict with the previous test results.
- Peng Y.X.; Chang X.D.; Zhu Z.C. Sliding friction and wear behavior of winding hoisting rope in ultra-deep coal mine under different conditions. Wear 2016, 368-369, 423-434.
- Chang X.D.; Peng Y.X.; Zhu Z.C. Effects of Strand Lay Direction and Crossing Angle on Tribological Behavior of Winding Hoist Rope. Materials 2017, 10(6), 1-20.
- Chang X.D.; Peng Y.X.; Zhu Z.C. Experimental investigation of mechanical response and fracture failure behavior of wire rope with different given surface wear. Int. 2018, 119, 208-221.
- Chang X.D.; Peng Y.X.; Zhu Z.C. Evolution properties of tribological parameters for steel wire rope under sliding contact conditions. Metals 2018, 8(10), 1-16.
- Chang X.D.; Peng Y.X.; Zhu Z.C. Effect of wear scar characteristics on the bearing capacity and fracture failure behavior of winding hoist wire rope. Int. 2019, 130, 270-283.
- Chang X.D.; Huang H.B.; Peng Y.X. Friction, wear and residual strength properties of steel wire rope with different corrosion types. Wear 2020, 458-459, 203425.
Finally, we appreciate for reviewer’s warm work earnestly, and hope that the corrections above will meet with approval.
Once again, thank you very much for your comments and suggestions.
Yours sincerely,
Qing Zhang

Round 2
Reviewer 4 Report
- The extreme pressure test is done with continuously increasing loads; which are the values of these loads? What was the seizure load?
Reply: In the extreme pressure test, the load value, namely the last non-seizure load value (PB), was obtained by setting the rotating speed, running time and temperature. Therefore, the load value does not appear in Table 1.
OK I found the results. However the results seem to be unrealistic. Normal values exceed 1000N and sometimes are as much as 6000N ! The values obtained must be undoubtedly explained !
- The values of the COF for IRIS (lubricant without additive) are greater than typical values obtained on similar tests; can the authors comment the obtained result, or eventually, compare it with some other results found in literature?
Reply: Thank you very much for the reviewer 's question. The CoF of IRIS is larger than that of LSMLO, and the reason for this phenomenon has been explained in this paper: Lamellar lanthanum stearate additive improves the anti-wear characteristics of IRIS.
My comment was about COF which is greater (for both additivated and non-additivated lubricant) than typical values obtained with many other lubricants (normally less than 0.1). The answer refers to "anti-wear characteristics" ?!
Undoubtedly, these tests seem to be more realistic. However, the test rig and the procedure lack a clear description. It is, at least, necessary to use citations referring to previously published papers of the authors.
Reply: The test rig in the following references are described in detail, so this paper only carried out a simple introduction, and the literatures are cited.
Here I expect a citation in the text; I could not find it.
- A comparison of these results with other obtained by the authors on the same test-rig seems to be necessary.
Reply: Several literatures have been published through this test-rig, and its repeatability can stand the test. The test conditions in this paper are different from the previous literatures. We think it is difficult to compare with the previous test results, but we have carried out repeated tests to verify the effectiveness of the results, and there is no conflict with the previous test results
The authors published a lot of papers with similar subjects so that is hardly acceptable that no comparison is possible.
Author Response
Response to Reviewer
Dear Reviewer:
Thank you for the comments concerning our manuscript entitled “Tribological Properties of Lanthanum Stearate Modified Lubricating Oil for Wire Rope in Mine Hoist” (No.: materials-1368121) again. Those comments are all valuable and very helpful for revising and improving our paper, as well as the important guiding significance to our research. We have studied comments carefully and have made correction which we hope meet with approval. Revised portion are marked in red in the revised manuscript. The main corrections in the paper and the responds to the reviewer’s comments are as follows:’’
- The extreme pressure test is done with continuously increasing loads; which are the values of these loads? What was the seizure load?
OK I found the results. However the results seem to be unrealistic. Normal values exceed 1000N and sometimes are as much as 6000N ! The values obtained must be undoubtedly explained !
Reply: We attach great importance to the question you raised, and conducted the experiments again. The same results were obtained on the four-ball rig in our laboratory according to the parameter standards set by us.
- The values of the COF for IRIS (lubricant without additive) are greater than typical values obtained on similar tests; can the authors comment the obtained result, or eventually, compare it with some other results found in literature?
My comment was about COF which is greater (for both additivated and non-additivated lubricant) than typical values obtained with many other lubricants (normally less than 0.1). The answer refers to "anti-wear characteristics" ?!
Reply: Thank you very much for the reviewer's question, and we apologize for the imperfect reply before. The larger CoF for IRIS may be related to equipment errors, human errors and the lubricating oil we selected. Here, we mainly compare the changes before and after adding additives.
Wire-rope sliding friction tests
Undoubtedly, these tests seem to be more realistic. However, the test rig and the procedure lack a clear description. It is, at least, necessary to use citations referring to previously published papers of the authors.
Here I expect a citation in the text; I could not find it.
Reply: Thank you very much to the reviewer. We have added the citation to the original text according to the reviewer’s requirement:
The anti-wear characteristics of LSMLO on hoisting wire ropes were evaluated by using the wire rope sliding wear test rig [16], as shown in Figure 2.
[16] Peng Y.X.; Chang X.D.; Zhu Z.C. Sliding friction and wear behavior of winding hoisting rope in ultra-deep coal mine under different conditions. Wear 2016, 368-369, 423-434.
- A comparison of these results with other obtained by the authors on the same test-rig seems to be necessary.
The authors published a lot of papers with similar subjects so that is hardly acceptable that no comparison is possible.
Reply: We are very sorry that the answer to this question did not satisfy the reviewer. Since the previous researchers did not use this test rig for experiments under lubrication conditions, we found a set of experimental data under dry friction as shown in the following figure. It can be seen from the figure that the CoFs of wire ropes also experience a rapid increase stage and a relatively stable stage, which are similar to the results obtained in our manuscript.
Fig. CoFs for different contact loads.(stroke:10mm).[16]
[16] Peng Y.X.; Chang X.D.; Zhu Z.C. Sliding friction and wear behavior of winding hoisting rope in ultra-deep coal mine under different conditions. Wear 2016, 368-369, 423-434.
Finally, we appreciate for reviewer’s warm work earnestly, and hope that the corrections above will meet with approval.
Once again, thank you very much for your comments and suggestions.
Yours sincerely,
Qing Zhang
